# Exosomal LncRNAs in Gastrointestinal Cancer: Biological Functions and Emerging Clinical Applications

**DOI:** 10.3390/cancers15030959

**Published:** 2023-02-02

**Authors:** Yuntong Sun, Fengtian Sun, Jianhua Jin, Wenrong Xu, Hui Qian

**Affiliations:** 1Jiangsu Key Laboratory of Medical Science and Laboratory Medicine, Department of Laboratory Medicine, School of Medicine, Jiangsu University, Zhenjiang 212013, China; 2Wujin Institute of Molecular Diagnostics and Precision Cancer Medicine of Jiangsu University, Wujin Hospital Affiliated with Jiangsu University, Changzhou 213017, China

**Keywords:** exosomes, lncRNAs, gastrointestinal cancer, biomarker, therapeutic target

## Abstract

**Simple Summary:**

Exosomes are nanosized vesicles secreted by living cells and exist in biological fluids. Exosomes contain various bioactive molecules, including proteins, nucleic acids, and lipids, which can reflect the pathophysiological state of donor cells. In recent years, long non-coding RNAs (lncRNAs) in exosomes have attracted wide attention in cancer research. Exosomal lncRNAs play important roles in gastrointestinal cancer progression. Blood samples-derived exosomal lncRNAs also exhibit great potential to serve as important indicators of liquid biopsy, thus providing a novel diagnosis strategy for gastrointestinal cancer. In addition, exosomes-mediated several lncRNAs delivery can exert anti-tumor effects. The aim of this review article is to summarize the recent work conducted in this field and evaluate the opportunities and challenges in utilizing exosomal lncRNAs for the diagnostic and therapeutic purposes of gastrointestinal cancer in the clinical setting.

**Abstract:**

Due to the lack of specific and effective biomarkers and therapeutic targets, the early diagnosis and treatment of gastrointestinal cancer remain unsatisfactory. As a type of nanosized vesicles derived from living cells, exosomes mediate cell-to-cell communication by transporting bioactive molecules, thus participating in the regulation of many pathophysiological processes. Recent evidence has revealed that several long non-coding RNAs (lncRNAs) are enriched in exosomes. Exosomes-mediated lncRNAs delivery is critically involved in various aspects of gastrointestinal cancer progression, such as tumor proliferation, metastasis, angiogenesis, stemness, immune microenvironment, and drug resistance. Exosomal lncRNAs represent promising candidates to act as the diagnosis biomarkers and anti-tumor targets. This review introduces the major characteristics of exosomes and lncRNAs and describes the biological functions of exosomal lncRNAs in gastrointestinal cancer development. The preclinical studies on using exosomal lncRNAs to monitor and treat gastrointestinal cancer are also discussed, and the opportunities and challenges for translating them into clinical practice are evaluated.

## 1. Introduction

Gastrointestinal cancer is one of the most common malignant tumors worldwide, accounting for 26% of the global cancer incidence and 35% of all cancer-associated deaths [1]. Currently, the diagnosis of gastrointestinal cancer mainly depends on digestive endoscopy and tissue biopsy [2]. However, these invasive methods may cause surgical trauma and infection and are not feasible for early screening and dynamic monitoring [3]. Several serum tumor markers, such as carcinoembryonic antigen (CEA), cancer antigen 19-9 (CA19-9), and alpha-fetoprotein (AFP), are also used to predict the risk of gastrointestinal cancer, whereas their sensitivity and specificity are still unsatisfactory [4]. Moreover, although many attempts have been made to improve the curative effect of cancer therapy, the treatment of many patients with advanced gastrointestinal cancer remains an important challenge. The major available treatments including surgery, chemotherapy, and radiotherapy may induce strong side effects and exhibit high recurrence rates [5]. Most patients are gradually resistant to drug therapy, resulting in the poor prognosis [6]. Therefore, it is of great significance to develop more effective strategies for the diagnosis and treatment of gastrointestinal cancer.

With the length of more than 200 nucleotides, a long non-coding RNA (lncRNA) is a type of ncRNAs that does not encode a protein [7]. Increasing studies have shown that lncRNAs display regulatory roles in various diseases using multiple mechanisms [8,9]. By inducing translational inhibition or functioning as competitive endogenous RNAs, lncRNAs are involved in the occurrence and development of gastrointestinal cancer [10,11,12]. Recent evidence has revealed that lncRNAs are the major component of exosomes, which are a subtype of extracellular vesicles (EVs) derived from living cells [13]. As a new manner of intercellular communication, exosomes-mediated lncRNAs delivery plays critical roles in many biological processes associated with gastrointestinal cancer, including tumor proliferation, metastasis, angiogenesis, drug resistance, and immune microenvironment. Several exosomal lncRNAs are also reported to be promising diagnostic and prognostic biomarkers and therapeutic targets for gastrointestinal cancer.

In this review, we overview the major characteristics and functions of exosomes and lncRNAs and summarize the biological roles and underlying molecular mechanisms of exosomal lncRNAs in gastrointestinal cancer progression. We also highlight the recent advances of exosomal lncRNAs as novel diagnostic biomarkers and therapeutic targets of gastrointestinal cancer and evaluate the future opportunities and challenges of this field.

## 2. Exosomes and lncRNAs

### 2.1. Biogenesis, Characteristics, and Functions of Exosomes

Because the specific markers of EV subtypes still need further investigations, the International Society for Extracellular Vesicles (ISEV) recommends researchers use operational terms for defining EVs based on physical characteristics (size and density), biochemical composition, or descriptions of conditions or cell of origin [14]. For example, EVs can be classified into small EVs (<100 nm or <200 nm) and medium/large EVs (>200 nm). In previous studies, EVs are usually divided into exosomes, microvesicles, and apoptotic bodies. However, strictly speaking, they should only be used when data showing the formation process are provided. In this review, in order to be consistent with the original article, we use the names of EVs that appear in the original article. Exosomes are the smallest subpopulation of EVs with a size ranging from 30–200 nm [15]. The production of exosomes involves several complex procedures, including endocytosis, endosome maturation, multivesicular bodies (MVBs) formation, and exosomes release [16,17]. Specifically, lipid raft domains of the plasma membrane induce the formation of early endosomes through the endocytosis pathway [18]. With the help of Golgi complex, these early endosomes gradually mature into late endosomes. Both endosomal sorting complex required for transport (ESCRT)-dependent and independent mechanisms are involved in MVBs formation [19]. After the fusion of MVBs with the plasma membrane, exosomes are finally released into the extracellular space. The internalization of exosomes by recipient cells includes three main ways: membrane fusion, receptor-ligand interaction, and endocytosis (Figure 1) [20].

Exosomes widely exist in human body fluids, such as blood, urine, semen, and saliva, and carry various bioactive molecules, including proteins, nucleic acids, and lipids, which can reflect the pathophysiological state of donor cells [21,22]. The lipid bilayer membrane structure endows exosomes with the natural property to protect cargos from degradation [23]. Although the exact mechanisms underlying the cargo sorting into exosomes remain unclear, several possibilities have been reported. The ESCRT pathway is recognized to mediate the introduction of proteins into MVBs in a ubiquitin-dependent manner [24]. The ESCRT system consists of four ESCRT subcomplexes and the ATPase vacuolar protein sorting-associated protein 4 (VPS4). Each component exerts unique functions sequentially, including cargo clustering (ESCRT-0), cargo binding (ESCRT-I and ESCRT-II), vesicle maturation and constriction (ESCRT-III), and membrane scission (VPS4) [25]. Moreover, several ESCRT-independent mechanisms also contribute to the secretion of exosomes. It is reported that ceramide, ADP ribosylation factor 6 (ARF6), and phospholipase D2 (PLD2) are associated with the cargo loading into exosomes [26]. The specific introduction of RNAs involves several important factors, including RNA binding proteins, adenylation, and urylation at the 3′ end of miRNAs, argonaute 2, and human antigen R, which can recognize RNAs with specific sequence and structure and then load them into exosomes [27].

Initially, exosomes were considered carriers to transport cell waste. Recent studies have shown that exosomes play critical roles in pathophysiological regulation. With high biocompatibility and stability, exosomes can be rapidly absorbed by recipient cells to transfer bioactive molecules [28]. By mediating intercellular communication, exosomes serve as homeostatic integrators in disease progression. Increasing evidence has revealed that exosomes derived from different cells display significant differences in regulating the behavior of recipient cells. For instance, stem cells-derived exosomes inherit the therapeutic potential of stem cells and exhibit repairing value in many diseases, such as diabetes, renal fibrosis, and spinal cord injury [29,30,31]. In contrast, exosomes from tumor cells are usually considered key players that promote tumor progression. Luo et al. demonstrate that intrahepatic cholangiocarcinoma cells release exosomes to upregulate programmed death-ligand 1 (PD-L1)-expressing macrophages, thus resulting in immune suppression and disease progression [32]. Exosomes have gradually become an important indicator of liquid biopsy because many components in exosomes can be remarkably changed with the development of disease.

### 2.2. Properties and Functions of lncRNAs

LncRNAs, a class of ncRNAs with the length of more than 200 nucleotides, are generated from the RNA polymerase II-mediated transcription and do not encode proteins [33]. According to the genomic origin, lncRNAs can be divided into five types: sense, antisense, bidirectional, intronic, and intergenic [34]. The mechanisms responsible for the synthesis of lncRNAs are not fully understood. In recent years, researchers have revealed that several key factors play important roles in the biogenesis of lncRNAs, including ribonuclease P-induced cleavage to produce mature ends, small nucleolar RNA-protein (snoRNP) complexes generation at their ends, and circular structures formation [35]. Emerging studies have shown that lncRNAs exist with high abundance in all organisms from prokaryotes to mammals [36]. There is an estimation of 5400 to more than 10,000 lncRNAs transcripts in humans [37]. The expression of lncRNAs in different tissues and cells exhibits high specificity [38]. With the development of diseases, the levels of lncRNAs are also changed significantly. The results of Alkhathami et al. indicate that patients with advanced breast cancer stage have elevated levels of lncRNA UCA-1 and HIT-1 compared to those with early-stage disease [39]. In addition, most lncRNAs are relatively conserved during the evolutionary progress [40]. These properties endow lncRNAs with great potential to serve as clinical biomarkers.

LncRNAs have been described as critical regulators in pathological and physiological processes. Through interacting with DNA, RNA, and proteins, lncRNAs are involved in the regulation of gene expression (Figure 2). The first role of lncRNAs is to regulate downstream gene transcription [41]. Under several stimulation conditions, lncRNAs can be specifically transcribed and participate in the transmission of signaling pathways. Chen et al. reveal that lncRNA HISLA inhibits the interaction of prolyl hydroxylase domain protein 2 (PHD2) and hypoxia-inducible factor-1alpha (HIF-1α) to stabilize HIF-1α expression, thereby promoting cancer aerobic glycolysis [42]. Increasing evidence has suggested that various lncRNAs can act as the sponge for miRNAs to attenuate their inhibition on downstream target genes [43]. For instance, lncRNA NEAT1 promotes the expression of myo-inositol oxygenase (MIOX) by competitively binding to miR-362-3p, thus increasing the sensitivity of hepatocellular carcinoma cells to ferroptosis [44]. LncRNA ITGB8-AS1 is reported to accelerate cell proliferation, colony formation, and tumor growth in colorectal cancer (CRC) by sponging miR-33b-5p and let-7c-5p/let-7d-5p to upregulate the expression of integrin alpha3 (ITGA3) and integrin beta3 (ITGB3) [45]. Moreover, lncRNAs are also considered as scaffolds, which enable several proteins to form complexes [46]. He et al. demonstrate that lncRNA MDHDH exhibits the ability to serve as a molecular scaffold for malate dehydrogenase 2 (MDH2) and 20S proteasomal core subunit alpha-type 1 (PSMA1), leading to the degradation of MDH2 [47]. The downregulation of MDH2 prevents the glycolysis progress in glioma cells. Zhou et al. suggest that lncRNA IFITM4P enhances PD-L1 expression to activate the immunosuppressive program in oral leukoplakia cells by facilitating the binding of SAM and SH3 domain containing protein 1 (SASH1) to phosphorylate transforming growth factor-beta-activated kinase 1 (TAK1) [48]. Another important function of lncRNAs is to combine with proteins to form complexes and then guide them to the specific DNA sequence, leading to the regulation of gene expression at the transcription level [49]. The results of Xiu et al. show that the binding of LINC02273 to heterogeneous nuclear ribonucleoprotein L (hnRNPL) activates anterior gradient homolog 2 (AGR2) expression by enhancing histone H3 lysine 4 trimethylation (H3K4me3) and histone H3 lysine 27 acetylation (H3K27ac) levels in the AGR2 promoter region [50].

## 3. Biological Functions of Exosomal lncRNAs in Gastrointestinal Cancer

Accumulating evidence indicates that lncRNAs are one of the major components of exosomes [51]. Several lncRNAs are enriched in exosomes. Through RNA sequencing, Wu et al. identify that 938 lncRNAs are highly expressed in ovarian endometriomas-derived exosomes [52]. Interestingly, lncRNAs in exosomes from different donor cells exhibit significant heterogeneity, implying that lncRNAs may be selectively sorted into exosomes [53]. Recently, exosomes-derived lncRNAs are reported to regulate the biological functions of recipient cells. The uptake of exosomes by recipient cells to deliver lncRNAs is regulated by many factors. The property of recipient cells may determine their interaction with exosomes [54]. The selective accumulation of exosomes in specific cells can affect the bioactive functions of exosomal lncRNAs [55]. As shown in Figure 1, exosomes transfer lncRNAs to receptor cells via three main processes: membrane fusion, endocytosis, and receptor-ligand interaction. The fusion of exosomes with the membrane of recipient cells is found to be activated with the decreased pH in the microenvironment [56]. Based on the specific cell type, exosomes can be absorbed by clathrin- or caveolae-mediated endocytosis, phagocytosis, or micropinocytosis [57]. Moreover, exosomes can also bind to the homologous receptors on the recipient cell membrane to trigger signal transduction reactions, thus transporting lncRNAs [58]. The natural plasma membrane structure of exosomes further enhances the stability of lncRNAs. Exosomal lncRNAs can be applied as diagnostic and prognostic biomarkers of various diseases [59]. Moreover, several disorders are considered to be associated with the dysregulation of lncRNAs [60]. Exosomes-mediated lncRNAs delivery represents a promising therapeutic strategy [61]. Therefore, lncRNAs have attracted widespread attention in exosome research.

Increasing studies have demonstrated that the occurrence and progress of gastrointestinal cancer are closely associated with the release of various exosomal lncRNAs. After their absorption by recipient cells, exosomes-mediated lncRNAs delivery regulates tumor proliferation, metastasis, angiogenesis, stemness, immune microenvironment, and drug resistance through multiple mechanisms (Figure 3 and Table 1).

### 3.1. Roles of Exosomal lncRNAs in Tumor Proliferation, Metastasis, Angiogenesis, and Stemness

During the occurrence and development of gastrointestinal cancer, exosomes serve as key mediators of intercellular communications. Emerging studies have suggested that exosomal lncRNAs are involved in all stages of tumor progression. Piao et al. demonstrate that lncRNA CEBPA-AS1 in gastric cancer (GC) cells-derived exosomes enhances the proliferation ability of tumor cells to facilitate GC carcinogenesis [62]. The results of Jin et al. reveal that lncRNA SND1-IT1 is transmitted to gastric mucosa cells by exosomes, leading to the malignant transformation of these cells by upregulating snail family transcriptional repressor 1 (SNAI1) expression [63]. In CRC, exosomes from cancer-associated fibroblasts (CAFs) transport lncRNA WEE2-AS1 to induce MOB kinase activator 1A (MOB1A) degradation and inhibit Hippo signaling pathway, thereby accelerating the growth of CRC cells [64]. Taniue et al. indicate that lncRNA UPAT promotes the survival and tumorigenicity of CRC cells by stabilizing ubiquitin-like with PHD and ring finger domains 1 (UHRF1) [65]. Moreover, increasing evidence has revealed that various cells-released exosomes promote the metastasis of gastrointestinal cancer by the delivery of lncRNAs. For instance, Fang et al. show that lncRNA PCAT1 derived from CRC exosomes promotes the tumor circulating cells-mediated CRC liver metastasis by regulating miR-329-3p/Netrin-1-CD146 axis [66]. LncRNA TTN-AS1, LINC01091, and FRLnc1 are enriched in GC cells-derived exosomes and exhibit the property to elevate the migration and invasion potential of GC cells [67,68,69]. The findings of Lian et al. demonstrate that the upregulation of lncRNA DUXAP10 is positively associated with advanced pathological stages, larger tumor sizes, and lymph node metastasis [70]. Epithelial to mesenchymal transition (EMT) is recognized as an important mechanism underlying tumorigenesis. Several exosomal lncRNAs are reported to exert critical roles in the EMT process of gastrointestinal cancer. GC cells exosomes-delivered lncRNA PCGEM1 can maintain the stability of SNAI1 to prompt the EMT of GC [71]. LncRNA TM4SF1-AS1 is able to promote the proliferation, invasion, and EMT of GC cell by activating PI3K/AKT signalling pathway [72]. LINC00659 is highly expressed in exosomes from CAFs and exosomal LINC00659 targets miR-342-3p to enhance the annexin A2 expression, thus activating EMT progress of CRC cells [73]. In addition, the angiogenesis and stemness of tumor cells are also found to be associated with many exosomal lncRNAs. Li et al. demonstrate that LINC01315 in CRC stem cells-derived exosomes enhances the stemness of CRC cells [74]. GC-secreted exosomal lncRNA X26nt inhibits VE-cadherin level to enhance vascular permeability and promote angiogenesis in a mouse subcutaneous tumor model [75].

### 3.2. Roles of Exosomal lncRNAs in Tumor Immune Microenvironment

Immune cells are the important components of tumor microenvironment and critically determine tumor progression. Zhi et al. report that BRAFV600E mutant CRC cells-derived exosomes can induce the formation of local immunosuppressive microenvironment by transmitting many lncRNAs, resulting in the peritoneal and distant lymph node metastasis [89]. These findings imply that exosomes from tumor cells may affect immune microenvironment. Recent studies have shown that exosomes-mediated lncRNAs delivery is involved in the modulation of various immune cells, such as macrophages, T cells, and natural killer (NK) cells. For instance, exosomes from MKN45 cells promote gastric carcinogenesis by lncRNA MIR4435-2HG-induced macrophage M2 polarization [76]. Neuroendocrine differentiated colon cancer cells-secreted exosomal lnc-HOXB8-1:2 enhances the migration and invasion capacities of tumor-associated macrophages through miR-6825-5p/CXC chemokine receptor 3 (CXCR3) axis [77]. Macrophages treated with CRC cells-derived exosomes exhibit increased M2 polarization to accelerate CRC metastasis, which is associated with the enrichment of lncRNA RPPH1 in exosomes [78]. Moreover, Xian et al. reveal that tumor cells-derived exosomes contribute to the immune escape of CRC cells by transporting lncRNA KCNQ1OT1 to suppress CD8^+^ T-cell response and induce PD-L1 ubiquitination [79]. Serum exosomes from CRC patients contain elevated lncRNA CRNDE-h level to activate Th17 cell differentiation by inhibiting the Itch-induced RORγt ubiquitination [80]. NK cells have a potent cytotoxic activity and exert important roles in host immune responses against tumor development [90]. The results of Huang et al. demonstrate that exosomal lncRNA SNHG10 from CRC cells inhibits the cytotoxicity of NK cells by enhancing inhibin subunit beta C (INHBC) expression, thus further facilitating CRC progression [81]. These findings indicate that exosomal lncRNAs represent novel messengers of tumor cells to regulate immune cell functions.

### 3.3. Roles of Exosomal lncRNAs in Drug Resistance

Although several anti-tumor drugs provide opportunities for gastrointestinal cancer therapy, drug resistance-induced unsatisfactory treatment effects remain a major challenge. Decreased drug uptake and increased drug efflux caused by many factors are responsible for the reduced drug effectiveness. A recent study reveals that exosomal lncRNA SNHG11 is highly expressed in bevacizumab-resistant CRC cells [82]. The knockdown of SNHG11 alleviates bevacizumab resistance and inhibits cell proliferation and migration, indicating that exosomal lncRNA can regulate the drug sensitivity in tumor cells. Moreover, exosomes-mediated crosstalk between cells is reported to endow tumor cells with drug resistance by transporting lncRNAs in the tumor microenvironment. Carcinoma-associated fibroblasts release exosomal lncRNA H19 and CCAL to activate the β-catenin pathway in CRC cells, thus accelerating tumor development and chemoresistance [83,84]. The enrichment of lncRNA CRNDE in M2-polarized macrophage-derived exosomes promotes the tumor growth in cisplatin-treated nude mice by downregulating phosphatase and tensin homolog (PTEN) expression in GC cells [85]. Increasing studies have suggested that various exosomal lncRNAs are involved in the dissemination of drug resistance from resistant cells to sensitive cells. For instance, CRC cells treated with exosomal lncRNA UCA1 derived from cetuximab-resistant cells exhibit increased cetuximab resistance [86]. Wang et al. demonstrate that lncRNA HOTTIP is upregulated in exosomes from cisplatin-resistant GC cells. HOTTIP promotes the cisplatin resistance in sensitive GC cells through the regulation on high mobility group A1 (HMGA1)/miR-218 axis [87]. Mitomycin-resistant CRC cells-derived exosomes-mediated transfer of lncRNA HOTTIP also contributes to the mitomycin resistance by modulating miR-214/karyopherin subunit alpha 3 (KPNA3) pathway [88]. These findings highlight the vital role of exosomal lncRNAs in drug resistance. The combined application of a specific lncRNA inhibitor and anti-tumor drug may improve the therapeutic effects on gastrointestinal cancer.

## 4. Exosomal lncRNAs as Potential Biomarkers of Gastrointestinal Cancer

Due to the lack of early symptoms, most patients with gastrointestinal cancer are diagnosed in the intermediate or terminal stage [91]. Researchers need to develop novel methods for the early diagnosis of gastrointestinal cancer to reduce the mortality. In recent years, exosomes are emerging as a major content of liquid biopsy and exist in biologic fluids, such as blood, urine, semen, and saliva [92]. During the occurrence and development of many diseases, exosomes contain specific bioactive molecules to reflect the pathological state of source cells [93]. The bilayer phospholipid structure also prevents the cargos from degradation. Although exosomes are a relatively smaller component of EVs, most lncRNAs display elevated levels in exosomes than that in apoptotic bodies and microvesicles [94]. The high abundance, stability, and specificity endow various exosomal lncRNAs with the potential to serve as biomarkers of gastrointestinal cancer.

Accumulating evidence has suggested that the circulating levels of exosomes-derived lncRNAs can help to predict the risk of tumor progression. Zhou et al. demonstrate that patients with GC have increased level of serum exosomal lncRNA H19 compared with healthy controls [95]. LncRNA NR038975, which is upregulated in serum exosomes from GC patients, promotes tumor cell proliferation, migration, and invasion by interacting with the NF90/NF45 complex [96]. The elevated lncRNA MIAT expression in serum-derived exosomes is closely associated with worse clinical variables and shorter survival, indicating that serum exosomal MIAT serves as an independent prognostic factor of GC [97]. Many attempts have been made to evaluate the clinical value of exosomal lncRNAs as diagnostic and prognostic biomarkers. Serum exosomal lncRNA GNAQ-6:1 from GC patients is expressed in low level and represents a new diagnostic biomarker with an area under ROC curve (AUC) value of 0.732 [98]. Exosomal lncUEGC1 is able to facilitate the detection of early GC patients from healthy individuals and those with premalignant chronic atrophic gastritis with great sensitivity and specificity [99]. The AUC values are 0.8760 and 0.8406, respectively. Circulating exosomal lncRNA-GC1 and HOTTIP present better diagnostic performance than CEA, CA72-4, and CA19-9 for distinguishing GC patients and healthy individuals [100]. Recent studies focus on the investigation of the relationship between exosomal lncRNAs and the tumor clinicopathological parameters. Pan et al. reveal that GC patients have elevated serum exosomal lncRNA ZFAS1 levels, which may contribute to the identification of lymphatic metastasis and TNM stage [101]. The results of Zheng et al. show that the increased expression of lncRNA SLC2A12-10:1 in plasma exosomes from GC patients is remarkably correlated with tumor size, TNM stage, lymph node metastasis, and differentiation degree [102]. Moreover, serum exosomal lncRNA PCSK2-2:1 is also considered as a potential diagnostic biomarker for GC due to the fact that the downregulation of PCSK2-2:1 can predict the tumor size, tumor stage, and venous invasion [103].

In CRC, several circulating exosomal lncRNAs are found to be differentially expressed. Liu et al. demonstrate that lncRNA GAS5 is upregulated in the tissues, plasma, and exosomes from patients with CRC [104]. Wang et al. reveal that serum exosomal lncRNA CCAT2 exhibits the ability to distinguish the CRC patients from the healthy controls [105]. Recently, many exosomal lncRNAs with great diagnostic potential have also been determined. The elevated level of serum exosomal lncRNA LINC02418 serves as a promising biomarker for the early diagnosis of CRC with an AUC of 0.8978 [106]. Yu et al. detect the serum exosomal lncRNA profiles of CRC patients and healthy individuals and reveal that lncRNA FOXD2-AS1, NRIR, and XLOC_009459 are remarkably upregulated in serum exosomes from CRC patients [107]. Notably, the combination of these three lncRNAs can achieve increased diagnostic value. Moreover, increasing studies have evaluated the correlation between exosomal lncRNAs and clinical data. CRC patients are reported to carry decreased exosomal lncRNA HOTTIP, which is significantly associated with poor overall survival [108]. Elevated levels of lncRNA 91H in serum exosomes from CRC patients show a high risk in tumor recurrence and metastasis [109]. In addition, exosomal lncRNA PVT1 contributes to the CRC metastasis by inhibiting miR-152-3p [110]. Overall, these findings indicate that exosomal lncRNAs are promising biomarkers of gastrointestinal cancer (Figure 4 and Table 2).

## 5. Exosomal lncRNAs as Therapeutic Targets of Gastrointestinal Cancer

Current treatment strategies for gastrointestinal cancer mainly rely on the chemotherapy-based cancer cell apoptosis, whereas the side effects and drug resistance are noteworthy [111]. With the revelation of their regulatory roles in tumor progression, exosomal lncRNAs may be developed as promising therapeutic targets. Li et al. report that the level of SPINT1-AS1 is increased in CRC tissues compared with adjacent normal tissues [112]. After surgical resection, CRC patients-derived serum exosomes exhibit reduced lncRNA SPINT1-AS1 level, implying that SPINT1-AS1 may be a molecular therapy target of CRC. Several approaches based on the function of exosomal lncRNAs for gastrointestinal cancer therapy have been reported. First, the exosomes-mediated delivery of lncRNAs with antitumor effects is considered a useful strategy. For instance, lncRNA PGM5-AS1 exhibits the ability to alleviate the proliferation, migration, and oxaliplatin tolerance of CRC cells. Treatment with PGM5-AS1 and oxaliplatin-loaded exosomes derived from 293T cells enhances the drug sensitiveness to improve the effectiveness of CRC therapy [113]. Li et al. also demonstrate that the overexpression of lncRNA ADAMTS9-AS1 inhibits cell proliferation and migration by downregulating Wnt/β-catenin signalling pathway, suggesting that engineered exosomes encapsulated with ADAMTS9-AS1 serve as a novel therapeutic agent of CRC [114]. In addition, various exosomal lncRNAs are found to promote the malignant behaviors of cancer cells. The inhibition of these exosomal lncRNAs may contribute to the suppression of gastrointestinal cancer. Yin et al. reveal that the expression of lncRNA NNT-AS1 is upregulated in serum exosomes from CRC patients [115]. The knockdown of NNT-AS1 significantly reduces the proliferation, migration, and invasion of CRC cells. Overall, the exosomal lncRNAs-mediated gastrointestinal cancer therapy is still in infancy; the safety and specific mechanism associated with this strategy need further investigations.

## 6. Challenges and Perspectives

Exosomal lncRNAs have attracted wide attention in cancer liquid biopsy and therapy due to their unique properties and important cellular functions. Increasing studies have revealed that exosomal lncRNAs are critically involved in the progression of gastrointestinal cancer by modulating tumor proliferation, metastasis, angiogenesis, stemness, immune microenvironment, and drug resistance. Although substantial breakthroughs have been made in the field of exosomes-mediated lncRNA delivery for tumor regulation, there still exist many challenges that may hinder the clinical application of exosomal lncRNAs: (1) the secretion mechanism of exosomal lncRNAs requires further explorations. Considering the heterogeneity of lncRNAs in exosomes derived from different source cells, future studies should focus on whether exosomes can package specifical lncRNAs during the formation process to participate in the gastrointestinal cancer progression. (2) Although several isolation methods of exosomes based on their physical and chemical characteristics have been established, the separation efficiency and the quality of exosomes are still unsatisfactory. Currently, most researchers adopt commercial exosome precipitation kits to purify exosomes from blood, whereas the purity of exosomes cannot meet the requirement of Minimal information for studies of extracellular vesicles 2018 [14]. (3) The rapid isolation and detection of exosomal lncRNAs with diagnostic value are still challenges. QRT-PCR is commonly used for the content analysis of exosomal lncRNAs. However, the sensitivity and accuracy of this method need further improvement. The analysis platforms and detection techniques for exosomal lncRNAs, such as digital PCR and loop-mediated isothermal amplification, may be promising candidates in clinical applications [116,117]. (4) Accumulating evidence has suggested that various serum/plasma exosomes-derived lncRNAs exhibit diagnostic potential in gastrointestinal cancer. The sensitivity, specificity, and AUC value of several exosomal lncRNAs are better than conventional tumor markers, such as CEA, CA19-9, and AFP. However, it is necessary to further expand the number of clinical samples and analyze the correlation with clinical data to validate the diagnostic effects of these exosomal lncRNAs. Notably, the combination of exosomal lncRNAs and tumor markers may achieve elevated diagnostic efficacy. (5) Exosomes-mediated therapeutic lncRNAs delivery has shown great value in the treatment of gastrointestinal cancer. There are still many practical problems to be solved before the clinical transformation of exosomal lncRNAs, including the route of administration, biodistribution, safety, biological functions, and effective dose. Moreover, the selection of the suitable source of exosomes should also be addressed.

## 7. Conclusions

In conclusion, exosomal lncRNAs exert critical roles in gastrointestinal cancer progression and exhibit great potential as novel diagnostic biomarkers and therapeutic targets. With the development of advanced technologies for the isolation and detection of exosomal lncRNAs, as well as the increasing studies on exosomal lncRNAs and gastrointestinal cancer, the resolution of these key issues will contribute to their clinical translation. We believe that exosomal lncRNAs-based diagnostic and therapeutic strategies can be effectively applied to clinical practice in the future.

## Figures and Tables

**Figure 1 cancers-15-00959-f001:**
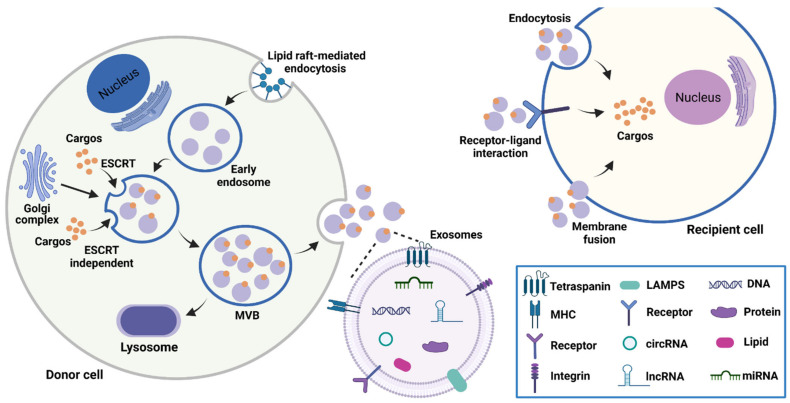
The biogenesis, contents, and uptake of exosomes. Exosomes are derived from the fusion of MVBs with plasma membranes. Exosomes transport cargos including nucleic acids, proteins, and lipids to recipient cells mainly through membrane fusion, receptor-ligand interaction, and endocytosis.

**Figure 2 cancers-15-00959-f002:**
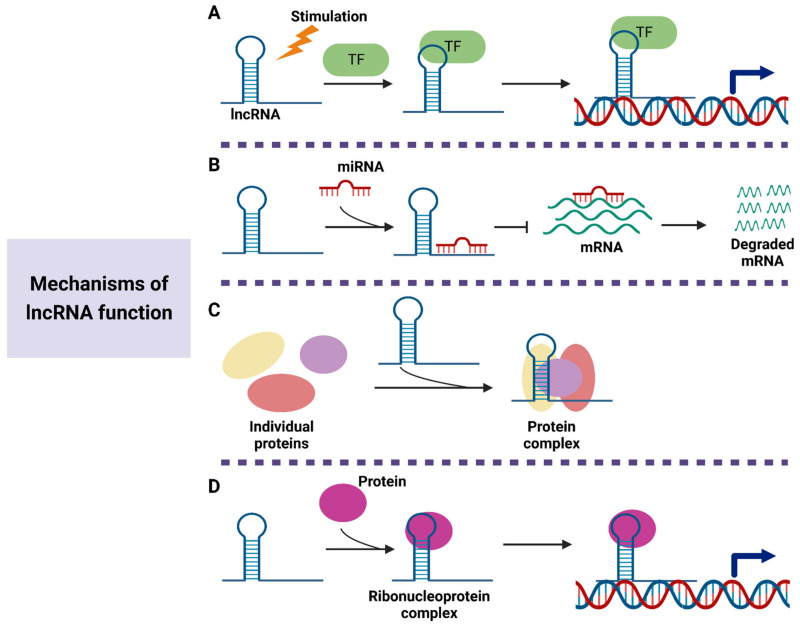
Functional mechanisms of lncRNAs. (**A**) Under several stimulation conditions, lncRNAs can combine with transcription factors to regulate signaling pathway. (**B**) LncRNAs prevent mRNA degradation by functioning as miRNA sponge. (**C**) LncRNAs can act as scaffolds to form protein complexes, thus regulating the transcription of the target genes. (**D**) LncRNAs modulate gene expression by guiding the ribonucleoprotein complex to the DNA sequence.

**Figure 3 cancers-15-00959-f003:**
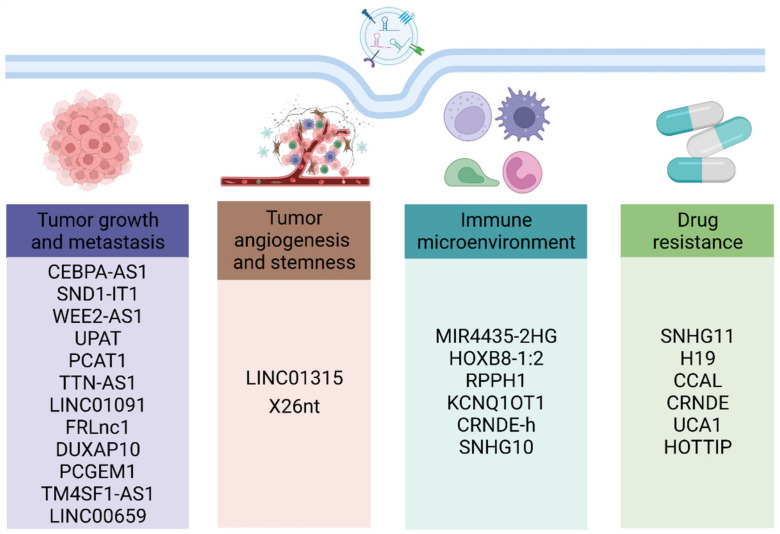
The biological role of exosomal lncRNAs in gastrointestinal cancer. Exosomal lncRNAs are involved in the various processes of gastrointestinal cancer including tumor growth, metastasis, angiogenesis, stemness, immune microenvironment, and drug resistance.

**Figure 4 cancers-15-00959-f004:**
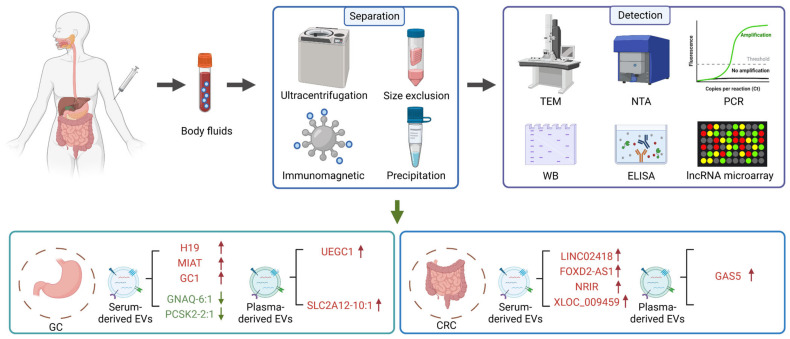
LncRNAs in EVs as new biomarkers for the liquid biopsy of gastrointestinal cancer. Several technologies have been established to isolate EVs from body fluids and detect their cargoes. Many lncRNAs in serum/plasma-derived EVs provide a new strategy for the diagnosis of gastrointestinal cancer. Note: ↑: Upregulation; ↓: Downregulation.

**Table 1 cancers-15-00959-t001:** The function and mechanism of exosomal lncRNAs in gastrointestinal cancer.

Cancer	lncRNA	Function	Mechanism	Reference
GC	CEBPA-AS1	Promoting proliferation and inhibiting apoptosis	N/A	[62]
GC	SND1-IT1	Inducing malignant transformation of GES-1 cells	MiRNA sponge for miR-1245b-5b to enhance USP3 expression	[63]
CRC	WEE2-AS1	Promoting proliferation	Inducing MOB1A degradation to inhibit Hippo pathway	[64]
CRC	UPAT	Promoting survival and tumorigenicity of CRC cells	stabilizing UHRF1 expression	[65]
CRC	PCAT1	Promoting colorectal cancer liver metastasis	MiRNA sponge for miR-329-3p to enhance Netrin-1 and CD146 expression	[66]
GC	TTN-AS1	Promoting growth and metastasis	MiRNA sponge for miR-499a-5p to enhance ZEB1 and CDX2 expression	[67]
GC	LINC01091	Promoting growth and metastasis	MiRNA sponge for miR-128-3p to enhance ELF4 and CDX2 expression	[68]
GC	FRLnc1	Promoting growth and metastasis	N/A	[69]
CRC	DUXAP10	Promoting growth and lymph node metastasis	Binding to LSD1 to inhibit the expression of p21 and PTEN	[70]
GC	PCGEM1	Promoting EMT process	Maintaining the stability of SNAI1	[71]
GC	TM4SF1-AS1	Promoting the proliferation, invasion and EMT	Activating PI3K/AKT signalling pathway	[72]
CRC	LINC00659	Promoting proliferation, migration, invasion and EMT progression	MiRNA sponge for miR-342-3p to enhance annexin A2 expression	[73]
CRC	LINC01315	Promoting proliferation, migration, and stemness	N/A	[74]
GC	X26nt	Increasing angiogenesis and vascular permeability	Reducing VE-cadherin expression	[75]
GC	MIR4435-2HG	Inducing macrophage M2 polarization to promote tumor growth	Activating the Jagged1/Notch and JAK1/STAT3 pathways	[76]
CRC	HOXB8-1:2	Inducing macrophage infiltration and M2 polarization to promote CRC progression	MiRNA sponge for miR-6825-5p to enhance CXCR3 expression	[77]
CRC	RPPH1	Inducing macrophage M2 polarization to promote the metastasis and proliferation of CRC cells	Inhibiting TUBB3 ubiquitination	[78]
CRC	KCNQ1OT1	Inhibiting CD8^+^ T-cell response to promote CRC cell immune escape	MiRNA sponge for miR-30a-5p to enhance USP22 expression	[79]
CRC	CRNDE-h	Activating Th17 cell differentiation to promote tumor growth	Inhibiting the Itch-mediated ubiquitination and degradation of RORγt	[80]
CRC	SNHG10	Suppressing the function of NK cells to promote CRC cell immune escape	Upregulating INHBC expression	[81]
CRC	SNHG11	Enhancing bevacizumab resistance in CRC cells	MiRNA sponge for miR-1207-5p to enhance ABCC1 expression	[82]
CRC	H19	Promoting the stemness and chemoresistance of CRC cells	MiRNA sponge for miR-141 to activate the β-catenin pathway	[83]
CRC	CCAL	Promoting chemoresistance of CRC cells	Interacting with HuR to enhance β-catenin expression	[84]
GC	CRNDE	Inducing cisplatin resistance in GC cells	Promoting NEDD4-1-mediated PTEN ubiquitination	[85]
CRC	UCA1	Promoting cetuximab resistance in CRC cells	N/A	[86]
GC	HOTTIP	Promoting cisplatin resistance in GC cells	MiRNA sponge for miR-218 to enhance HMGA1 expression	[87]
CRC	HOTTIP	Increasing resistance of CRC cells to mitomycin	MiRNA sponge for miR-214 to enhance KPNA3 expression	[88]

Abbreviations: ABCC1: ATP-binding cassette subfamily C member 1; CDX2: caudal-type homeobox-2; CRC: colorectal cancer; CXCR3: CXC chemokine receptor 3; ELF4: E74-like factor 4; EMT: epithelial–mesenchymal transition; HMGA1: high mobility group A1; INHBC: inhibin subunit beta C; JAK1: janus kinase-1; KPNA3: karyopherin subunit alpha 3; LSD1: lysine-specific demethylase 1; GC: gastric cancer; GES-1: gastric epithelial cells; MOB1A: MOB kinase activator 1A; N/A: not applicable; NEDD4-1: neuronal precursor cell-expressed developmentally downregulated 4-1; NK: natural killer; PTEN: phosphatase and tensin homolog; SNAI1: snail family transcriptional repressor 1; STAT3: signal transducer and activator of transcription 3; TUBB3: beta-III tubulin; UHRF1: ubiquitin-like with PHD and ring finger domains 1; USP22: ubiquitin specific protease 22; USP3: ubiquitin specific protease 3; ZEB1: zinc finger E-box binding homeobox 1.

**Table 2 cancers-15-00959-t002:** Diagnostic significance of exosomal lncRNAs in gastrointestinal cancer.

Cancer	lncRNA	Expression	Source	Case Number	AUC	Reference
GC	H19	Upregulation	Serum-derived EVs	81	0.849	[95]
GC	MIAT	Upregulation	Serum-derived EVs	109	N/A	[97]
GC	GNAQ-6:1	Downregulation	Serum-derived EVs	43	0.732	[98]
GC	UEGC1	Upregulation	Plasma-derived EVs	10	0.876 (distinguish GC patients from healthy individuals) 0.8406 (distinguish GC patients from chronic atrophic gastritis patients)	[99]
GC	GC1	Upregulation	Serum-derived EVs	826	0.9033	[100]
GC	SLC2A12-10:1	Upregulation	Plasma-derived EVs	120	0.776	[102]
GC	PCSK2-2:1	Downregulation	Serum-derived EVs	63	0.896	[103]
CRC	GAS5	Upregulation	Plasma-derived EVs	158	0.964	[104]
CRC	LINC02418	Upregulation	Serum-derived EVs	155	0.8978	[106]
CRC	FOXD2-AS1	Upregulation	Serum-derived EVs	203	0.728	[107]
CRC	NRIR	Upregulation	Serum-derived EVs	203	0.660	[107]
CRC	XLOC_009459	Upregulation	Serum-derived EVs	203	0.682	[107]

Abbreviations: AUC: area under ROC curve; CRC: colorectal cancer; GC: gastric cancer; N/A: not applicable.

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
