# Peer review of "Exosomal LncRNAs in Gastrointestinal Cancer: Biological Functions and Emerging Clinical Applications"

_cancers, 2023, doi:10.3390/cancers15030959_

Round 1

Reviewer 1 Report

Dear editor,

This review entitled Exosomal LncRNAs in Gastrointestinal Cancer: Biological 2Functions and Emerging Clinical Applications” summarize the recent work conducted in this field and evaluate the opportunities and challenges in utilizing exosomal lncRNAs for the diagnostic and therapeutic purposes of gastrointestinal cancer in the clinical setting. However, there are a few point could be revise

1.       According to the recommendation of ISEV2018, exosomes can be replaced by EV or sEV, please refer to the recommendations of ISEV2018 in detail, and quote and amend them in the review.

2.       In figure4, ’plasma derived EV”\ serum derived EV, this definition is more appropriate

3.       LNC carrying EV uptake can also be included in the review.

Author Response

Jan 29, 2023

Cancers

Manuscript ID: cancers-2182570

Title: Exosomal LncRNAs in Gastrointestinal Cancer: Biological Functions and Emerging Clinical Applications

Authors: Yuntong Sun†, Fengtian Sun†, Jianhua Jin, Wenrong Xu* and Hui Qian*

Dear Editor,

Thank you for your e-mail dated Jan 26, 2023 regarding the review of our manuscript. We appreciate your assessments of the manuscript and have found that the comments and suggestions are helpful in preparation of the revised manuscript. We revised the manuscript as suggested by reviewers. In the revised manuscript, all the amendments were highlighted by yellow words. The following are our point-by-point responses, in order of the comments about the manuscript:

Reviewer 1

Dear editor,

This review entitled “Exosomal LncRNAs in Gastrointestinal Cancer: Biological Functions and Emerging Clinical Applications” summarize the recent work conducted in this field and evaluate the opportunities and challenges in utilizing exosomal lncRNAs for the diagnostic and therapeutic purposes of gastrointestinal cancer in the clinical setting. However, there are a few point could be revise

1. According to the recommendation of ISEV2018, exosomes can be replaced by EV or sEV, please refer to the recommendations of ISEV2018 in detail, and quote and amend them in the review.

Response: Thanks for your suggestion. We have refered to the recommendations of ISEV2018 in detail, and quoted and amended them in our revised manuscript.

2: In figure4, ’plasma derived EV”\ serum derived EV, this definition is more appropriate

Response: Thanks for your suggestion. We have corrected it in our revised manuscript.

3: LNC carrying EV uptake can also be included in the review.

Response: Thanks for your suggestion. We have added a description about the uptake of EVs carring lncRNAs in Section 3.

We hope that the above responses meet the expectations of the reviewer. The comments and suggestions are helpful in improving the quality of our manuscript.

Thank you for your consideration. We look forward to hearing from you and to publication of this manuscript.

Sincerely,

Hui Qian, Prof.

School of Medicine, Jiangsu University,

301 Xuefu Road, 212013, Zhenjiang, Jiangsu, P.R. China.

E-mail: 1000007341@ujs.edu.cn

Reviewer 2 Report

This review deals with lncRNAs in gastrointestinal cancer and addresses their potential for diagnostics and therapeutics.

The manuscript is well described and may help to understand the role of lncRNAs in its pathophysiology and the search for novel targets for biomarkers. It is scientifically sound and contains sufficient interest and originality to merit publication.

Minor comments:

1. Some genes have already been reported to be associated with GI cancers but are not listed in Fig3 or Table 1. For example, UPAT and DUXAP10 in colorectal cancer and TM4SF1-AS1 in gastric cancer. Please consider that discussions about them could be added.

Author Response

Jan 29, 2023

Cancers

Manuscript ID: cancers-2182570

Title: Exosomal LncRNAs in Gastrointestinal Cancer: Biological Functions and Emerging Clinical Applications

Authors: Yuntong Sun†, Fengtian Sun†, Jianhua Jin, Wenrong Xu* and Hui Qian*

Dear Editor,

Thank you for your e-mail dated Jan 26, 2023 regarding the review of our manuscript. We appreciate your assessments of the manuscript and have found that the comments and suggestions are helpful in preparation of the revised manuscript. We revised the manuscript as suggested by reviewers. In the revised manuscript, all the amendments were highlighted by yellow words. The following are our point-by-point responses, in order of the comments about the manuscript:

Reviewer 2

This review deals with lncRNAs in gastrointestinal cancer and addresses their potential for diagnostics and therapeutics.

The manuscript is well described and may help to understand the role of lncRNAs in its pathophysiology and the search for novel targets for biomarkers. It is scientifically sound and contains sufficient interest and originality to merit publication.

Minor comments:

1. Some genes have already been reported to be associated with GI cancers but are not listed in Fig3 or Table 1. For example, UPAT and DUXAP10 in colorectal cancer and TM4SF1-AS1 in gastric cancer. Please consider that discussions about them could be added.

Response: Thanks for your suggestion. We have added the citations about these lncRNAs in the section 3, Figure 3 and Table 1 in our revised manuscript.

We hope that the above responses meet the expectations of the reviewer. The comments and suggestions are helpful in improving the quality of our manuscript.

Thank you for your consideration. We look forward to hearing from you and to publication of this manuscript.

Sincerely,

Hui Qian, Prof.

School of Medicine, Jiangsu University,

301 Xuefu Road, 212013, Zhenjiang, Jiangsu, P.R. China.

E-mail: 1000007341@ujs.edu.cn

Reviewer 3 Report

Dear Authors,

congratulations for the excellent Review you wrote. Current knowledge on the exosomal LncRNAs in gastrointestinal cancer, in terms of biological functions and future clinical appications has been clearly described and is supported by a robust bibliography.

Author Response

Jan 29, 2023

Cancers

Manuscript ID: cancers-2182570

Title: Exosomal LncRNAs in Gastrointestinal Cancer: Biological Functions and Emerging Clinical Applications

Authors: Yuntong Sun†, Fengtian Sun†, Jianhua Jin, Wenrong Xu* and Hui Qian*

Dear Editor,

Thank you for your e-mail dated Jan 26, 2023 regarding the review of our manuscript. We appreciate your assessments of the manuscript and have found that the comments and suggestions are helpful in preparation of the revised manuscript. We revised the manuscript as suggested by reviewers. In the revised manuscript, all the amendments were highlighted by yellow words. The following are our point-by-point responses, in order of the comments about the manuscript:

Reviewer 3

Dear Authors,

congratulations for the excellent Review you wrote. Current knowledge on the exosomal LncRNAs in gastrointestinal cancer, in terms of biological functions and future clinical appications has been clearly described and is supported by a robust bibliography.

Response: Thanks for your positive comments. We appreciate your assessments of the manuscript.

We hope that the above responses meet the expectations of the reviewer. The comments and suggestions are helpful in improving the quality of our manuscript.

Thank you for your consideration. We look forward to hearing from you and to publication of this manuscript.

Sincerely,

Hui Qian, Prof.

School of Medicine, Jiangsu University,

301 Xuefu Road, 212013, Zhenjiang, Jiangsu, P.R. China.

E-mail: 1000007341@ujs.edu.cn
